# A Case Study of Tourist Perceptions and Revisit Intention Regarding Qingwan Cactus Park in Penghu, Taiwan

**Paul Juinn Bing Tan** [1,*] **, Hui-mei Yao** [1] **and Ming-Hung Hsu** [2]

1 Department of Applied Foreign Languages, National Penghu University of Science and Technology, Magong 880, Taiwan; jessyhm@gms.npu.edu.tw
2 Department of Electrical Engineering, National Penghu University of Science and Technology, Magong 880, Taiwan; hsu@gms.npu.edu.tw
* Correspondence: tanjuinnbing@gmail.com

**Abstract:** In recent years, the citizens of Taiwan have exhibited an increasing demand for domestic tourism and travel options. Due to their relatively early development as a domestic tourism destination, the Penghu Islands are well known for their rich natural and cultural resources. The purpose of this study was to examine factors influencing tourists' visits to Qingwan Cactus Park, including their reasons for visiting and their perceptions of their visits. With principal component analysis serving as the basis for an IPA methodology, and using the visiting destination as a reference point, the results obtained can provide a basic guideline for tourism planning. The visitors' reactions and demands were re-examined through IPA analysis. The results could be used by national park managers to develop constructive suggestions on implementing sustainable development in Taiwan's tourism industries. This study presents valuable data on Penghu and provides analyses of residents' and tourists' opinions, as well as their reactions. The study's conclusions can be extrapolated to research on other destinations outside of Taiwan.

**Keywords:** Penghu (Pescadores); (IPA) interpretative phenomenology analysis; forward expectations; practical experience; tourist perceptions; revisit intention; qualitative research method approach; importance-performance analysis (IPA)



## 1. Introduction

The notion of sustainability has been applied in various fields since its conception. Young (1992) [1] pointed out that sustainability has been introduced into a wide range of disciplines, giving rise to concepts such as sustainable ecology, sustainable economy, sustainable agriculture, sustainable community development, sustainable human development, etc. Even though sustainability is emphasized differently in each discipline, the implications of sustainable development remain more or less the same. However, it is still widely held that a comprehensive sustainable development plan should simultaneously entail the three dimensions of environment, economy, and society, as these three dimensions depend on and mutually influence one another and should be taken into consideration. Ultimately, sustainable development should reach an optimum condition where all three systems can be sustained continuously. Many scholars have advocated that in the pursuit of environmental sustainability, the methods, systems, information, or finances utilized should not lead to the depletion of natural resources or destruction of natural life cycles [2,3]. Social cultural sustainability emphasizes the positive effects of economic efficacy on the environment and society. Meanwhile, productive efficiency and allocative efficiency are evaluation factors that promote sustainable businesses and economies. Productive efficiency is the point of optimal production where marginal cost is equal to marginal benefit and waste production is prevented. Allocative efficiency refers to the allocation of resources such that the situation will improve for some people, but not at the expense of others.

Tourism has emerged as an attractive option to achieve economic growth and regional development in several rural areas. Indeed, the beneficial effects of rural tourism on economy have been under study for a significant amount of time [4,5]. Tourism can diversify the economies of rural communities, especially communities located on islands that are small and difficult to access.

Without a doubt, be it for natural landscapes, cultural industries, or local food businesses, the utilization of advertisements, marketing strategies, and promotions are all attempts to expedite tourism growth in the region, and thereby generate more benefits. This study is focused on Penghu, an outlying island region of Taiwan that has been a longstanding focus and direction of the nation's tourism industry.

The Penghu Islands possess rich resources for tourism and recreational activities, including not only twisting stretches of coastline and other natural ecological resources but also many historical monuments which are worth visiting and preserving.

Penghu's Qingwan Cactus Park is one of the notable sights in Penghu, and it is located close to many other important tourist attractions, such as Shili Beach, Fonggui Cave, and Shetou Hill, among others. Therefore, the park and the surrounding area comprise a must-see attraction for tourists. With its beautiful landscapes featuring different types of cactus vegetation, as well as a wide offering of recreational and educational activities and multi-functional facilities, the park has brought about considerable prosperity and development to the area.

As defined by the World Tourism Organization (WTO), which in turn extracted the definition from the Brundtland Report, sustainable tourism consists of tourism that "meets the needs of tourists and host regions." Despite debates over the concept, the above definition provides a clear path to follow [6,7] (Clarke, 1997, Hardy et al., 2002). For this reason, there is a need to formulate tourism policies with the objective of safeguarding natural, social and cultural resources, and to guarantee that the management of these resources fulfills the necessities of both actual and future residents and tourists [8,9] (Sharpley, 2000, Liu, 2003, Blancas' et al., 2010).

In adherence to this new paradigm, the tourism planning policies aspire to develop a model of tourism characterized by quality, sustainability and diversity that can enhance the competitive capacity of destinations [10].

In the study "The assessment of sustainable tourism: Application to Spanish coastal destinations" (Blancas et al., 2010), the WTO proved to be a guide that provides indicators of sustainable development in terms of tourism destinations [11] (World Tourism Organization, 2004). These indicators of sustainable tourism can be determined and subdivided into the three subsets, namely key and complementary indicators of sustainable tourism, and specific site indicators. Blancas et al., (2010) described how a sustainable tourism indicator system's information can be defined, quantified, and used for application in Spanish coastal destinations. The results of their analysis can serve as a practical basis for defining, quantifying, and utilizing data obtained from other sustainability indicators applied to other destinations for the green industries worldwide.

This study focused on visitors to Qingwan Cactus Park to assess whether travel expectations and tourist motivations may alter their perceptions towards the park as well as their intentions to revisit. Perception scales are useful tools for measuring people's views regarding products, work situations, their quality of life, community characteristics, and the quality of outdoor recreation options, among other things. In seeking to understand and satisfy the needs of tourists, it is most important to explore their motivations and perception levels and, in order to understand these factors, it is necessary to take appropriate measurements.

### 1.1. Purposes of This Study

- Understand the value of tourist perceptions and provide professional management knowledge to enhance the management quality of national park services.
- Provide useful assessment of sustainable tourism for national park sightseeing.

- Determine the factors that contribute to Penghu visitors' intentions and behaviors with regard to revisiting the cactus park.
- Provide the local government with a better decision-making process.
- Judge whether the satisfaction of tourists' demands in terms of the recreational quality of national park services plays a significant role in Penghu.

### 1.2. Literature Review

Tourism accounts for more than 8% of the foreign exchange revenue earned globally, and the total revenue figure exceeds those of other forms of international trade. Generating around US$5.33 trillion in revenue, tourism is the most dominant form of international trade. Moreover, tourism is listed among the top five sources of income in 83% of countries around the world and contributes to 38% of the foreign exchange revenue of certain countries. For island countries or developing countries/regions, mass tourist numbers is a key driver of the tourism industry as well as related economies. Undoubtedly, tourism constitutes a pivotal green industry in the 21st century. Therefore, tourism has been and will continue to be a key player in economic development worldwide, together with aiding the development of economies in individual countries. Many scholars have regarded tourism as the world's greatest green industry to date [12].

In connection with this research subject, this study investigated the factors of forward expectations, practical experiences, motivations, perceptions, and revisit intentions among tourists. The following sections provide a review, discussion, and brief compendium of pertinent literature to make the theoretical basis of the study clear and better explain the overall structure of this study.

### 1.3. Forward Expectations Theories and Implications

The term "expectation", at least as used in the present research, was defined by Parasuraman, Zeithaml, and Berry (1988) [13] to refer to the requirements and desires of customers. These three scholars believe that what a provider should offer is far more important than what can be provided by a provider. In his theory of expectation, Lawler (1973) suggested that people regard their abilities to attain different achievements with different preferences, and engage in related actions accordingly. That is, people carry certain psychological expectations regarding their own abilities [14].

Miller (1977) demonstrated that expectations could be classified into four different types: "ideal", "expected", "minimum tolerable", and "desirable" expectations [15]. Iso-Ahola, S. E. & Allen J. R. (1982) viewed the recreation demands and motivations of tourists as often being subject to the influence of personal characteristics and their past experiences [16]. Due to people potentially having different expectations regarding various forms of recreation, with these expectations leading in turn to their recreation-related behaviors, there are also environmental qualities, types of activities, and real situations that cause different recreational experiences, with these factors in turn eventually affecting a given tourist's recreational perceptions (Figure 1).

To summarize, the aforementioned scholars generally agree that people regard their abilities to attain different achievements with different preferences. Moreover, people may have different expectations regarding recreational activities that lead to different recreational behaviors, which in turn cause different recreational experiences that affect their recreational perceptions.

### 1.4. Practical Experiences and Relevant Research

The process of a recreational experience includes five stages: a phase of anticipation phase, followed by the actual travel phase, then the on-site-activities phase, followed by the return travel phase and, finally, the recollection phase [17,18]. Hammitt (1980) explained that when engaging in recreational activities, a person's senses, satisfactions, mind, and behavior will bring about continuous interactions with various environmental factors that will in turn yield various feelings and experiences. Such experiences are called recreational

experiences [18]. According to Y. L. Chao & S.Y. Chao (2017), "Experience is individual that after it subject to external stimulation, through the emotion and satisfactions process to come about physiology and psychological reaction, and can be categorized into two groups according to the type of stimulation: intrinsic and extrinsic" [19,20].

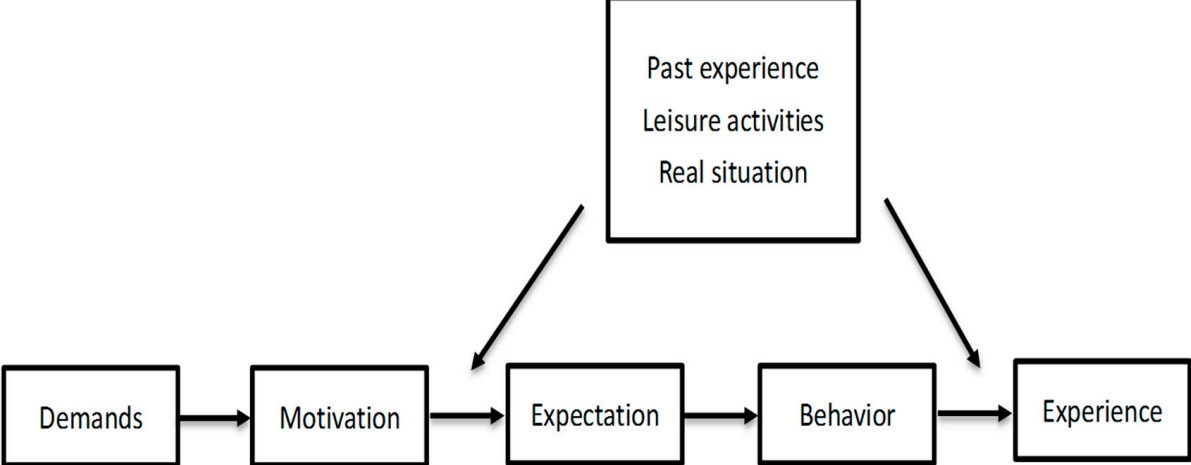

**Figure 1.** Tourists Recreational Perceptions.

Recreational degree of recognition refers to tourists visiting and reacting to a variety of natural and cultural landscapes, recreation facilities, security facilities, service qualities, and so on. Recreational perceptions refers to the subjective evaluations of tourists regarding recreational activities and experiences, which are affected by various subjective and objective factors during those recreational activities [21].

The concept of a recreational experience and its relevant phases are shown in Figure 2.

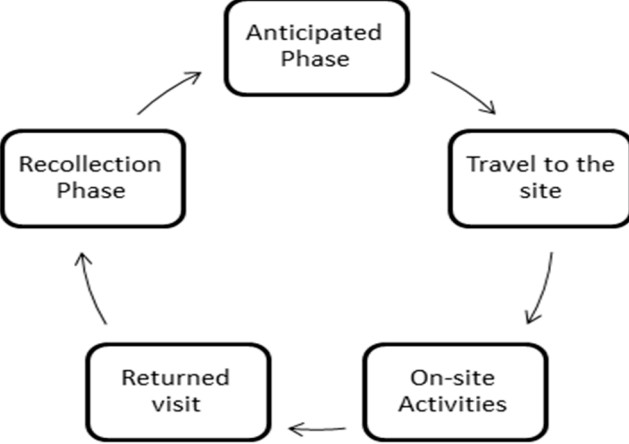

**Figure 2.** Recreational Experience and its relevant phases.

To summarize, the aforementioned scholars generally agree that when joining in recreational activities, a person will have continuous interactions with various environmental factors that will in turn yield various feelings and experiences. These feelings and experiences essentially comprise the subjective evaluations of tourists regarding recreational activities and experiences.

### 1.5. Tourist Motivations

The motivation to travel is a "drive". This drive not only compels people to travel to meet social needs and mental needs but is a major factor in what recreational activities tourists choose to take part in (Iso-Ahola & Allen, 1982) [16]. When people engage in

tourism activities, they have different tourist motivations. These motivations in turn result in behaviors that, if effective, bring the person to question perceptions [22].

Wu, H. C., Huang, C. T., & Hung, C. M. (2015) pointed out "the travel behavior of tourists" is controlled by both long-term motivations and short-term motivations. When people get together on weekends after a busy week or take a short trip to a destination close to home, they are generally driven by short-term motivations. However, when people plan a travel itinerary and collect related information for several months before taking a trip, they are being driven by long-term motivations [23].

Tourism is a high-quality leisure activity that allows people to reduce their stress levels and avoid over-stimulation in a relaxing quiet environment. Individuals can express themselves through leisure travel, and such travel will typically improve an individual's physical and spiritual well-being. When a whole family travels together, the relationships among the family members will be improved and become harmonious through their shared participation in leisure activities. For this reason, tourism makes people fully relaxed, prepared for their next tasks, and more efficient in their work [23].

In recent years, the needs and development of domestic tourism and leisure recreation have been deemed as substantial elements that influence social and economic development [24]. According to the Oxford English Dictionary definition of "psychology": "motivation and behavior are two relative constructs. The individual behavior was outside activity; on the other hand, the motivation was internal process promoted by individual action". However, the majority of psychologists agree that: ''the motivation not only caused and maintains individual activities but also was an internal process, and promoted activities achieved the goal" [25].

A growing body of literature has revolved around discussing tourist motivations. Crompton (1979) pointed out that the factors influencing the objectives of tourism destination choices are called tourist motivations, which can be categorized into pull and push factors [26]. Among the seven main push factors are: escaping from reality, surveying self-esteem, relaxing, seeking prestige, recovery and promotion of family relationships, and advancing interpersonal relationships. The two major pull factors are newfangled objects and education. Thomas (1964) proposed that there are eighteen kinds of important travel motivations, which he grouped into in four major categories in 1964. [27] (Table 1).

**Table 1.** Travel Motivations.

| Motivation | Explanation |
|---|---|
| Recreation & Entertainment | To break away from one's job and the duties of daily routine. To play and have a good day. To go and obtain romantic experiences with the opposite sex. |
| Education and Culture | Not only to see how the locals work, live, and entertain themselves, but also to go sightseeing and attend special festivals. |
| Ethnic traditions | To pay respects to forebears. To visit any locations where family and friends have been to visit. |
| Others | Others refers to motivations related to climate, health, sports, the economy, and adventure. These motives are related to the ability to surpass others, to the pursuit of fashion, to participating in history, and to having a desire to understand the world. |

In conclusion, the motivation to travel is a "drive". This drive not only compels people to travel to meet social needs and mental needs but is a major factor in what recreational activities tourists choose to take part in. Individual people can express themselves through leisure travel, and when a whole family travels together, relationships among the members will be improved and become harmonious through their shared participation in leisure activities.

*1.6. Tourist Satisfaction and Perceptions*

"Satisfaction is one of the personality traits that an individual possesses, as an attitude, a feeling, an indistinct and abstract noun". With regard to satisfaction, scholars have a lot of definitions of the word. Kotler (1997) took the view that customers have a tally board in

their mind to determine whether they are satisfied or dissatisfied after purchasing goods or being served. The higher customer satisfaction is regarding a product or service, the better the chances a client will purchase that product or service again [28].

According to Dorfman (1979) [21], recreational satisfaction refers to a tourist's overall evaluation of a recreational experience, while Martin (1988) regarded overall satisfaction as a measurement of the total sum of recreational expectations at different levels and the recreational experiences that have been obtained. Martin (1988) stated that there should be consistency between an individual's expectations towards a certain experience and the actual experience that he or she gains during a trip. An individual will have a feeling of satisfaction if what he or she experiences is equal to or exceeds what he or she expected prior to the trip. Otherwise, an individual will not have a feeling of satisfaction with what he or she experiences.

Recreational quality, which is considered an element of the recreational experience, makes reference to the level of tourist needs met by recreational opportunities and the resulting satisfaction. If the manager of a destination provides recreation opportunities that better satisfy a tourist's demands, the satisfaction in terms of the recreational quality will be higher [23].

The satisfaction of tourists is important to site managers. Tourist attractions that result in higher tourist satisfaction will have higher rates of sales and revisit intentions. Therefore, high tourist satisfaction is something that managers must strive to achieve. Hung et al., (2005) proposed a measurement of satisfaction in a study regarding the cultural tourism development of the Tai-An Railroad that includes four dimensions and environmental features with eleven variables (Table 2).

**Table 2.** Dimensions and Environmental Features.

| Items | Measured by | |
|---|---|---|
| Psychological experience | 1. | air factors |
| | 2. | cultural characteristics |
| | 3. | relationships |
| Recreation environment | 1. | environmental characteristics |
| | 2. | construction of environment |
| Recreational facilities and services | 1. | public facilities |
| | 2. | service quality |
| | 3. | participation of residents |
| Recreational activities | 1. | exhibition and selling of special local products |
| | 2. | active function |
| | 3. | guided tours |

In summary, the aforementioned scholars agree that when a person has an experience, he or she will have expectations that are met, not met, or exceeded by the actual experience. When those expectations are either fulfilled or the result is better than anticipated, the customer will be satisfied. If not, they will not be satisfied. In other words, if the manager of a destination provides recreation opportunities that better satisfy a tourist's demands, the satisfaction in terms of the recreational quality will be higher.

*1.7. Revisit Intentions*

Revisit intention refers to whether tourists are willing to travel again to a given destination or to other scenic spots in the same country [29]. In this study, we defined "revisit intention" as tourists being willing to travel to a single destination or a spot again. There have been numerous studies that have investigated tourists' satisfaction with various attractions. Therefore, such studies have made some relevant contributions to the relevant domestic and foreign literature.

Lai et al., (2018) reported that tourists' overall satisfaction positively correlates with marine activities and their intentions to revisit [30–33], while Huang et al., (2005) found

that revisit intentions are based on customer introductions, public praise, and word of mouth [31,33].

### 1.8. Current Literature on Destinations

The concept of tourism sustainability entails implementing controls to monitor the carrying capacity of a certain destination. This matter of carrying capacity in particular has drawn increasing interest from academia and the public institutions behind tourism planning. Published by the World Tourism Organization (WTO), the Global Code of Ethics for Tourism proposes that the different actors implicated in tourism ought to take into account their agenda while also considering the interests of the community, together with the site's natural environment and cultural patrimony. (Antonio Alvarez-Sousa, Jose Luis Paniza Prados, 2020) [34,35]. A survey by Turistes titled "Tourists Barcelona 2016" was conducted between 22 February and 22 December 2016 and enrolled a total of 6032 tourists (over 15 years old) that stayed for between 1 and 28 nights in Barcelona. The CAPI computer system was utilized to carry out the interviews and data was analyzed using the program Atlas.ti, after which the grounded theory phases and the functions in Atlas.ti were integrated. The questionnaire items used for assessing the perception of tourists in terms of carrying capacity in Barcelona were as follows: 38_2 (representing the degree of agreement or disagreement with "Barcelona is too crowded for tourism") and 38_3 (representing the degree of agreement or disagreement with "In Barcelona prices are higher than the quality offered"). The SPSS software was used to analyze the data [36–39] (Antonio Alvarez-Sousa, 2018).

### 1.9. Study by Antonio Alvarez-Sousa and Jose Luis Paniza Prados on "Visitor Management in World Heritage Destinations before and after COVID-19, Angkor"

This particular study focused on analyzing both visitor-management tactics and strategies implemented in World Heritage destinations, utilizing as case studies the Cambodian temples of Angkor. Even though the sustainability paradigm was maintained, including risk society concepts and public health objectives as central elements was an important complement [35,40,41].

In short, the aforementioned scholars agree that a statistically significant positive relationship exists between overall satisfaction in tourists and marine activities and their revisit intentions.

The Penghu Islands are a chain of islands to the west of the main island of Taiwan. In high season, the islands offer many popular and dynamic activities; for example, playing water sports, experiencing a fisherman's life, and visiting historical and cultural landmarks. On the other hand, Penghu also has many kinds of tourist attractions that can be further developed in the future, such as Qingwan Cactus Park, which contains hundreds of cacti and various facilities. As such, the park definitely offers many fun and nature-related experiences to visitors.

In order to accommodate tourists seeking to visit the privately run Qingwan Cactus Park in Penghu, it is also necessary to market Penghu itself as a beautiful tourist destination. Conducting this research can benefit the Penghu County administration and assist Qingwan Cactus Park in turning itself into a must-visit place, as well as one of the famous scenic spots in Penghu [42,43].

This research study focused on forward expectations, practical experiences, tourist motivations, tourist satisfaction, and tourist revisit intentions as five major factors, as well as on related theories and literature [12,44]. The study was intended to investigate current tourists' opinions of and impressions of Qingwan Cactus Park.

Previous research has claimed that what moves people to travel and, simultaneously, what lures them to visit certain destinations is simply motivation [45,46]. In the past, several studies were conducted in order to pinpoint the push and pull factors that contribute to forge tourists' motivations, in addition to elucidating how to utilize these factors in the creation of more effective marketing strategies [26,45–50] (see Table 3).

**Table 3.** Previous research on the underlying factors that determine revisit intentions.

| Author(s) | Theory or Model | Year | Destination/Country |
|---|---|---|---|
| Antonio Alvarez-Sousa & Angkor, Jose Luis Paniza Prados [35] | grounded theory | 2020 | Cambodia |
| Christina K., Leonidas H., Thomas F., Dimitrios F [42] | market segmentation push and pull motivation | 2020 | Greece |
| Jun Wen & S (SAM) Huang [51] | push and pull motivation | 2019 | Cuba |
| Antonio Alvarez-Sousa [39] | grounded theory | 2018 | Barcelona, Spain |
| Senutha P Ratthinan & Nor Hafizah Selamat [52] | Push and pull motivation | 2017 | Penang, Malaysia |
| Rojan Baniya & Kirtika Paudel [53] | Push and pull motivation | 2016 | Nepal |
| N.V.Seebaluck, P.R.Munhurrun, P.Naidoo, P.Rughoonauth [54] | Push and pull motivation | 2015 | Mauritius |
| Mai Ngoc Khuong & Huynh Thi Thu Ha [55] | Push and pull motivation | 2014 | Ho Chi Min City, Vietnam |
| lan Phau, Sean Lee, Vanessa Quintal [49] | Push and pull motivation | 2013 | Australia |
| Baahar A.M., A.H. Mohammad & Ahmad P. M. Som [56] | Push and pull motivation | 2010 | Jordan |

A number of previous studies have also emphasized social media relevance in terms of offered services and products. Since experience is a main contributor to the perception of travel service, visitors consider that online reviews are a high-value source of information [57], to the point where these reviews exert considerable influence on the demand for a tourist service. Accordingly, Collie reported that, of all the travel organization made by travelers beforehand, around 69% is defined by online reviews obtained from previous tourists [42]. Whereas online reviews for sightseeing or activities at a certain destination are harder to find, those for restaurants or hotels are more frequently available. Despite them all being experiences related to tourism, they can be considered services [57]. Therefore, there is a necessity for tourism-related marketing to recognize the role of social media in terms of travelers' perception of a destination [49,50].

## 2. Materials and Methods

The current study's main purpose was to investigate the effects of tourists' expectations and experiences on their levels of satisfaction with the Qing wan Cactus Park in Penghu. To do so, the study collected data on the satisfaction of actual visitors to the park. It was initially hypothesized that the expectations of park visitors and the quality of the services provided would have a direct association with the visitors' overall satisfaction with their visits. In a general sense, the research aimed to determine the applicability of satisfaction-related theories to the specific case of the Qingwan Cactus Park in Penghu.

### 2.1. Participants

The participants in this study consisted of park visitors who viewed the park as a tourist attraction to be visited during a trip to the islands. More specifically, the study surveyed a total of 413 tourists who came to the park. These 413 tourists were randomly selected from among all the visitors to the park during a certain period and included individuals from a range of different cities.

This study further explores the methodology proposed by Oh et al. [47], which proposes classifying tourists into diverse market segments according to their travel motivation. This segmentation contributes to predicting the motivation behind destination choices in travelers. This study aims to examine whether the segmentation analysis using push and

pull factors could be applicable to the context of social media. In particular, we assess whether the motives of tourists to travel are influenced by their preferences during the information-seeking phase prior to travel within the social media context. The following research challenges are addressed by this study (See Figure 3).

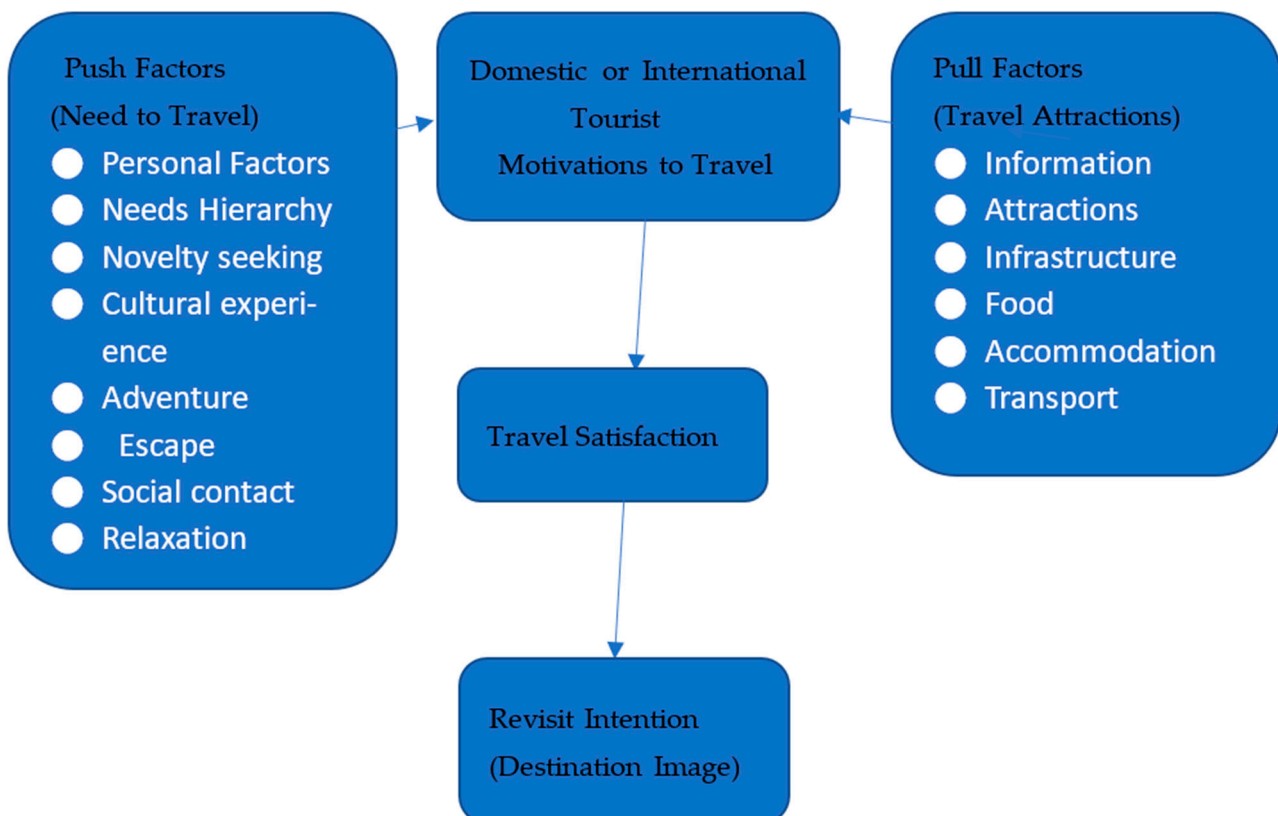

**Figure 3.** Research Model. Tourism-related Push and Pull Factors: A Theoretical Model theory and Model [26,58].

### 2.2. Instruments

The selected tourists filled out a questionnaire that was a modified version of a formerly used survey. The questionnaire consisted of two separate sections in relation to two central factors: satisfaction and revisit intention. Four main topics were included in the sections referring to expectations and satisfaction as follows: (1) Public facilities, (2) BBQ areas, (3) Leisure and recreation facilities, and (4) Leisure and recreation areas. Throughout the questionnaire, the respondents were asked to rate their level of agreement with numerous elements using a 5-point Likert Scale, in which travelers can express their agreement or disagreement ranging from 1 = strongly disagree to 5 = strongly agree. To analyze visitors' perceptions, we employed several approaches, including interpretative phenomenology analysis, IPA, and qualitative research methods.

### 2.3. What Is IPA?

With an IPA approach [9], a study's advantageous elements increase fourfold as this approach allows researchers to bond with the study participants and form a relationship. In addition, due to its qualitative nature, IPA constitutes an ideal opportunity for researchers to better acquaint themselves with the innermost deliberation of "lived experiences" from interviewees (study participants). Being a "participant-oriented" approach, the interpretative phenomenological analysis approach makes it possible for interviewees to express themselves and convey their "lived experience" stories in a manner that they deemed to be appropriate and in the absence of any distortion and/or prosecution. Smith et al. (2009) [59] established that the field of psychology gave origin to IPA, as well as a considerable number of previous studies that utilized IPA were health psychology

studies. Since then, IPA has seen increasingly frequent use, particularly in clinical and counseling psychology, as well as social and educational psychology. Unsurprisingly, the core constituent of IPA is, broadly speaking, applied psychology, that is, psychology in a real world setting. Being a tool especially designed for qualitative research, IPA enables numerous individuals with similar experiences to re-tell their stories in an environment without any distortions and/or prosecutions. Creswell (2012, p. 76) affirmed that [60], "a phenomenological study describes the common meaning for several individuals of their lived experiences of a concept or phenomenon". With respect to research population and sample size, Creswell (2012) [60] explained that determining the size of the sample that will be required is vital when selecting study participants. Similarly, a typical phenomenological study can have a participant sample size of between 2 and 25. Participants in an IPA research study should also be drawn from a homogeneous sample pool, for which Smith et al. (2009) [59] pointed out the main challenge in IPA is the following: "IPA is a detailed account of individual experience. The issue is quality, not quantity, and given the complexity of most human phenomena, IPA studies usually benefit from a concentrated focus on a small number of cases". A phenomenological research study conducted by Creswell (2013) [61] revealed that collecting information is a process that entails primary in-depth interviews with up to 10 participants, in which the key is to describe the essence of the phenomenon for the few who have experienced it. Therefore, collecting data for an IPA study should proceed following these instructions: IPA research should include semi- and non-structured interviews, where between two (2) and twenty-five (25) individuals participate.

Components of IPA research. At its core, the phenomenological research approach (i.e., IPA) grants the capacity to explore, evaluate, and understand the research participants' "lived experiences". Creswell (2013) [61] asserted that the exploratory capacity in qualitative research enables investigating, interpreting, and understanding the problems encountered in these studies. Additionally, he claimed that "We conduct qualitative research because a problem or issue needs to be explored"; therefore, he was of the opinion that the phenomenological approach is most suitable for uncovering the root-cause of a phenomenon.

### 2.4. Procedures

The developed questionnaire was distributed to the randomly selected visitors to the park from August 2019 to September 2019. In other words, the study's measurements of tourists' perception came from data collected from tourists when they were still at the park. This approach was taken because it meant that the tourists would be in the midst of experiencing their visits and, thus, would have fresh and clear opinions regarding the facilities, attractions, and customer services of the park. The total number of questionnaires distributed was 500, of which a total of 413 returned questionnaires were valid, which represents an effective rate of 82.6%. The questionnaire was intended to measure the visitors' levels of satisfaction with various aspects of the different facilities/areas in the park.

### 3. Results

*3.1. Characteristics of Respondents*

There were 230 (55.7%) male and 183 (44.3%) female respondents among the total of 413 people who took part in this research. Approximately 234 (56.7%) of the visitors were 21 to 30 years old, while 29.1% of the visitors were 13 to 20 years old. With respect to education level, 51.6% of the respondents had a college education, 6.3% of the respondents had graduated from a senior high school, 6.1% of the respondents had received specialized subject education, and 6.5% of the respondents had a graduate school education or above. As such, more than 50% of the respondents had a high level of education. In terms of occupation, 67.1% of the respondents were students, while 10.2% were in the service industry and 5.1% worked in the fields of business. With respect to income, 52.5% of the respondents earned less than NTD 10,000 per month, and 17.2% earned NTD 10,001–20,000 per month.

Most of the respondents were unmarried (88.1%) and lived in Penghu City (54.0%), and most had come to Qingwan Cactus Park with friends or colleagues (54.7%).

### 3.2. Tourist Motivations

Tables 4 and 5 summarize the tourist motivations of the visitors. According to the strength of their motivations, the specific motivations were scored from 0 to 5. Over half of the respondents were motivated by wildlife appreciation (3.68), sightseeing (3.67), relaxing (3.59), improving friendships (3.26), getting out of the city (3.24), and just visiting while passing by (3.12).

**Table 4.** Tourist motivations (Push Factors).

| Factors (Push Motivators) | Mean | Standard Deviation |
|---|---|---|
| Sightseeing | 3.67 | 0.98 |
| Relaxing | 3.59 | 1.04 |
| Improving Friendship | 3.26 | 1.19 |
| Getting out of the city/Escape | 3.24 | 1.25 |
| Historical site/Cultural experience | 2.47 | 1.30 |
| Learning about environmental Protection | 2.85 | 1.28 |
| Wildlife Appreciation | 3.68 | 0.94 |
| Exercise and Fitness | 2.48 | 1.32 |
| Photography | 2.80 | 1.44 |
| Social events | 2.75 | 1.33 |
| Satisfying Curiosity | 3.37 | 1.11 |
| Work or Educational Purpose | 2.23 | 1.44 |
| Recommendation from family and friends | 2.83 | 25 |
| Achievement | 2.28 | 1.40 |
| Just visiting while passing by | 3.12 | 1.18 |
| Visit (spontaneous) | 2.93 | 1.18 |
| Stargazing | 2.01 | 1.44 |
| Memorable Experience | 1.97 | 1.48 |

**Table 5.** Tourist motivations (Pull factors).

| Factors (Pull Motivators) | Mean | Standard Deviation |
|---|---|---|
| Sightseeing | 3.26 | 1.19 |
| Getting out of the city | 3.24 | 1.25 |
| Historical site | 2.47 | 1.30 |
| Learning about environmental Protection | 2.85 | 1.28 |
| Wildlife Appreciation | 3.68 | 0.94 |
| Exercise and Fitness | 2.48 | 1.32 |
| Photography | 2.80 | 1.44 |
| Social events | 2.75 | 1.33 |
| Satisfying Curiosity | 3.37 | 1.11 |
| Work or Educational Purpose | 2.23 | 1.44 |
| Recommendation from family and friends | 2.83 | 1.25 |
| Achievement | 2.28 | 1.40 |
| Just visiting while passing by | 3.12 | 1.18 |
| Visit (spontaneous) | 2.93 | 1.18 |
| Stargazing | 2.01 | 1.44 |
| Memorable Experience | 1.97 | 1.48 |

### 3.3. Forward Expectations

The expectations of the participants are detailed in Table 6. We can see that the minimum and maximum values of the forward expectations were 3.02 and 4.25, respectively (out of a possible maximum of 5). The average value was 3.66. Therefore, only 1/3 of the values were over the average. Those expectation values included the expectation values for restrooms (4.25), shaded spaces (4.17), parking lots (4.05), recreation spaces (4.02), medical

facilities (4.00), lakeside terraces (3.85), lighting facilities (3.83), posters (3.81), and display facilities (3.68). The tourists' top three expectations requiring satisfaction at the Qingwan Cactus Park were therefore those for restrooms (4.25), shaded spaces (4.17), and recreation spaces (4.02).

**Table 6.** Forward expectations (X).

| Items (24) | Mean | Standard Deviation |
|---|---|---|
| (1) Tourist visitor center | 3.61 | 1.09 |
| (2) Signage | 4.02 | 0.92 |
| (3) The trail beside basalt | 3.53 | 0.95 |
| (4) Recreation spaces | 4.02 | 0.85 |
| (5) Shaded spaces | 4.17 | 0.88 |
| (6) Restaurants | 3.64 | 1.08 |
| (7) Parking lots | 4.05 | 0.91 |
| (8) Restrooms | 4.25 | 0.86 |
| (9) Lakeside terraces | 3.85 | 0.99 |
| (10) Dream hall | 3.38 | 1.17 |
| (11) Greenhouse cultivation | 3.60 | 1.02 |
| (12) Dahsianhua place | 3.54 | 1.03 |
| (13) The area where animals are found | 3.56 | 1.10 |
| (14) The place where there are many lover locks | 3.02 | 1.30 |
| (15) Rock flower place | 3.26 | 1.09 |
| (16) Chienhuhua place | 3.25 | 1.14 |
| (17) Lighting facilities | 3.83 | 1.11 |
| (18) Medical facilities | 4.00 | 1.12 |
| (19) Historic sites | 3.15 | 1.25 |
| (20) Explanation of handbook | 3.63 | 1.11 |
| (21) Explanation of billboard | 3.81 | 0.99 |
| (22) Explanation of poster | 3.52 | 1.07 |
| (23) Explanation of multimedia | 3.44 | 1.16 |
| (24) Facilities Display | 3.68 | 1.12 |

### 3.4. Practical Experiences (Tourist Satisfaction)

The practical experiences of the tourists indicated the degree to which their expectations were satisfied. In this regard, as indicated in Tables 6 and 7, the maximum and minimum values were 2.8 and 3.48, respectively (out of a possible maximum of 5). The average value was 3.13. Regarding all the questions, the responses for only a few questions were over the average of 3.13. The results revealed that most of the visitors' responses were "disagree" or "neutral", which caused the average value to fall near the middle of the scale. That is, the visitors were not that satisfied with the performances of the factors they were asked about. The results also indicated that most of the respondents had a bad impression of Qingwan Cactus Park from their visit and were also not satisfied with these factors after their visit.

### 3.5. Importance-Performance Analysis (IPA)

In order to understand the visitors' reactions to the questions, we decided to design a detail-oriented questionnaire. At the same time, we used IPA to analyze every average of "importance" and "performance". Second, we put the results on a two-dimensional coordinate graph, and then described the performance of the items in order to have the priorities of the information provided. It was by using this analysis that we could easily know where the items fell in different quadrants. As shown in Figure 4, 10 items (41%) fell in quadrant C, which means that the "importance" and "performance" of those items were of low priority. In contrast, seven items (29%) were located in quadrant A, which showed that those items should maintain the good work. Taken together, the results discussed above suggested that Qingwan Cactus Park visitors were less likely to be satisfied by revisiting.

**Table 7.** Practical experiences (Y).

| Items (24) | Mean | Standard Deviation |
|---|---|---|
| (1) Tourist visitor center | 3.07 | 1.11 |
| (2) Signage | 3.42 | 0.94 |
| (3) The trail beside basalt | 3.24 | 1.06 |
| (4) Recreation spaces | 3.38 | 1.02 |
| (5) Shaded spaces | 3.29 | 1.09 |
| (6) Restaurants | 2.87 | 1.25 |
| (7) Parking lots | 3.36 | 1.05 |
| (8) Restrooms | 3.48 | 1.09 |
| (9) Lakeside terraces | 3.39 | 1.07 |
| (10) Dream hall | 3.15 | 1.09 |
| (11) Greenhouse cultivation | 3.31 | 1.04 |
| (12) Dahsianhua place | 3.23 | 1.09 |
| (13) The area where animals are found | 3.13 | 1.14 |
| (14) The place where there are many lover locks | 2.94 | 1.25 |
| (15) Rock flower place | 3.06 | 1.15 |
| (16) Chienhuhua place | 3.04 | 1.16 |
| (17) Lighting facilities | 3.07 | 1.22 |
| (18) Medical facilities | 2.87 | 1.37 |
| (19) Historic sites | 2.92 | 1.29 |
| (20) Explanation of handbook | 2.87 | 1.31 |
| (21) Explanation of billboard | 3.24 | 1.10 |
| (22) Explanation of poster | 2.95 | 1.26 |
| (23) Explanation of multimedia | 2.80 | 1.32 |
| (24) Facilities Display | 3.12 | 1.21 |

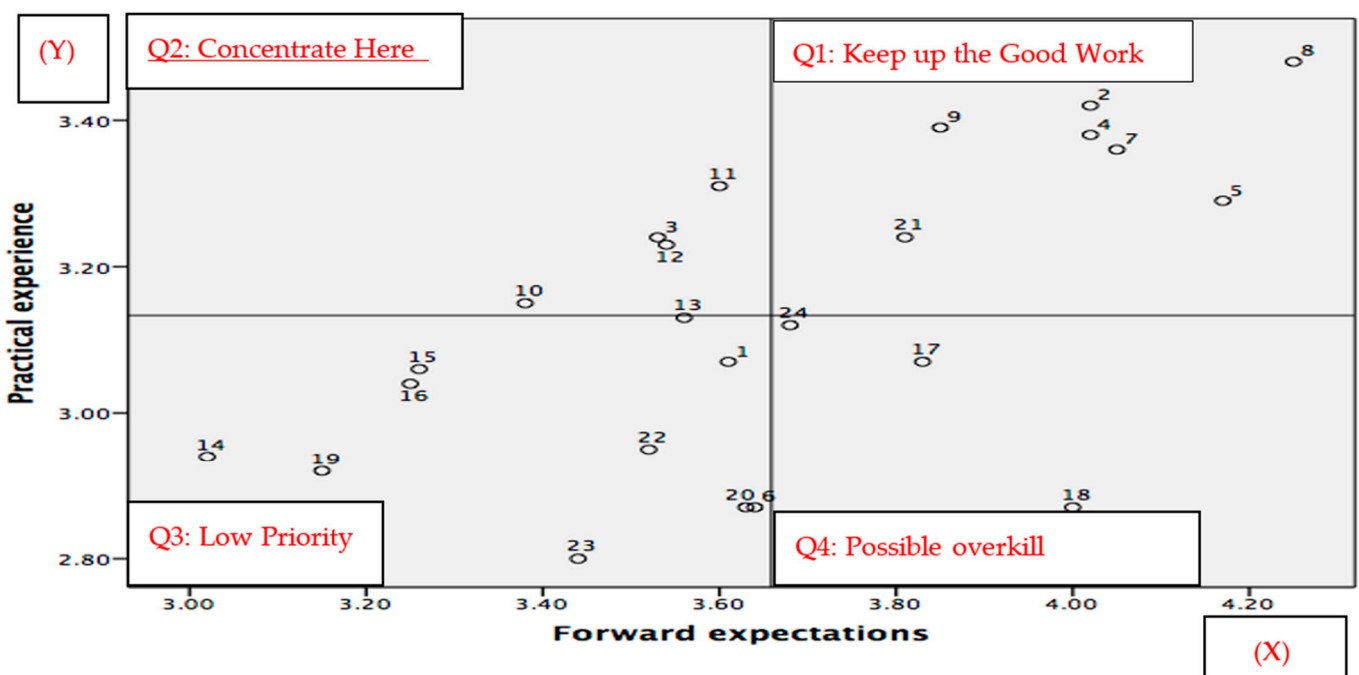

**Figure 4.** Quadrant diagram of IPA Four Quadrants of IPA matrix.

*3.6. Revisit Intentions*

The revisit intentions of the visitors are shown in Tables 8 and 9. There were 36.3% (29.5 + 6.8%) of respondents who were satisfied with the park. The percentage of respondents who were willing to revisit and spread the word about the park was 35.1% (26.6 + 8.5%). However, if the entrance ticket price was to be increased, the proportion of respondents indicating a willingness to revisit was reduced to 20.6%. A quarter of

the respondents were neutral about encouraging their friends to come to the park, but 41.4% of the respondents would do so. Only 27.4% (19.9 + 7.5%) of the respondents said they would view the park as a priority travel destination in the future, while 39.0% of the respondents were neutral and 27.8% of respondents said they would not view it as a priority travel destination.

**Table 8.** Averages of forward expectations and practical experiences.

|  | N | Minimum | Maximum | Mean | Std. Deviation |
|---|---|---|---|---|---|
| Forward expectations | 24 | 3.02 | 4.25 | 3.6588 | 0.32409 |
| Practical experiences | 24 | 2.80 | 3.48 | 3.1333 | 0.20071 |
| Valid N (listwise) | 24 |  |  |  |  |

**Table 9.** Perceptions of revisit intentions and overall satisfaction of tourists.

| Items | Mean | Standard Deviation |
|---|---|---|
| Overall satisfaction with Qingwan Cactus Park | 3.23 | 0.93 |
| Intention to revisit Qingwan Cactus Park | 3.09 | 1.09 |
| If the entrance ticket price is raised higher in the future, your intention to revisit | 2.55 | 1.21 |
| I'll spread the advantages of Qingwan Cactus Park to others | 3.08 | 1.13 |
| I'll encourage my relatives and friends to experience the park | 3.17 | 1.10 |
| Qingwan Cactus Park will be a priority travel destination in the future | 2.79 | 1.26 |

## 4. Discussion

Indeed, Penghu has emerged among domestic visitors as a major, highly-popular tourist attraction. Compared with other scenic areas in Penghu, Qingwan Cactus Park is relatively new and features well-equipped facilities, including interactive and educational programs. The Park contains both natural and cultural elements, such as various plants, animals and historic monuments, as well as areas devoted to various do-it-yourself (DIY) activities. Since its opening in 2015, Qingwan Cactus Park has gradually become a popular and must-visit place.

The participants in the study consisted of visitors to the park who were invited to complete a questionnaire. Although the total number of questionnaires distributed was 500, only 413 of the returned questionnaires were valid, yielding an effective rate of 82.6%. After collecting the questionnaires, the SPSS statistical software was used to analyze the results, with the aim of determining future trends for Qingwan Cactus Park through the analysis of various factors, including tourist motivations, tourist perceptions, and revisit intentions.

In fact, planning and management cannot occur without the participation of different actors and particularly citizens; in this respect, the typologies proposed by Arsntein [62] and Tosun [63,64] are particularly helpful to identify (a) the opportunities provided by the government structure (b) their corresponding levels with regard to a specific moment. In the same vein, Mitchell's creative destruction theory [65–69] can serve as a useful tool for understanding the state of a conflict at specific moment and the interventions that could follow.

The World Tourism Organization's (WTO) definition was drawn from the Brundtland Report and defines sustainable tourism as the type of tourism that "meets the needs of tourists and host regions, while ensuring and enhancing future opportunities" [10].

This study analyzed and compared the differences in the surveyed visitors' satisfaction levels prior to and after their practical experiences. We used IPA (Importance-Performance Analysis) to analyze the visitors' satisfaction levels. From the data shown in Tables 6–9, we can see that the top three features required for satisfaction included restrooms (4.25), shaded spaces (4.17), and recreation spaces (4.02). Relatedly, Qingwan Cactus Park is located in the south of Penghu, and it has the only public restroom on the road. We were impressed by this result, which exceeded our expectations. Furthermore, shaded spaces and recreation spaces were expected by visitors of every age. That was because of the

scorching sun in summer and the windy weather in winter, which made people have the intention of finding a place for rest or shade.

Otherwise, the visitors also made some suggestions about Qingwan Cactus Park. For example, some said, "There should be more advertising for this place so that it can attract more visitors to go sightseeing. Also, it needs to fully use its own characteristics, such as using the basalt to crush out the sparks with cactus. However, now there are too many large buildings so that the original advantage can't be seen by people". Others said, "The signs are unclear, making it hard for visitors to know where to go". Apparently, the park needs to take all these issues into consideration, and we hope that this study can help the park managers to understand its strengths and weaknesses so as to make necessary changes.

Our study is a source for national park managers and policymakers, along with communication strategists, since it presents crucial practical and managerial implications. These should be taken into consideration to better understand how push and pull factors can influence the intentions of tourists to revisit a destination, as well as the behavioral processes behind choosing a national park as a destination.

When it comes to our analysis, it should be noted that the participants may have lacked actual observations or personal experiences of using some of the facilities in the park; thus, they may not have formed specific opinions regarding some of the questions. Therefore, it may be assumed that they had no opinions toward certain questions when they filled out the questionnaire. Relatedly, the expectations of the participants were personal opinions regardless of their actual observations or experiences. This may have resulted in the higher validity of the participants' expectations. This may explain the greater difference between the average and maximum values and the intermediate values in terms of the participants' expectations rather than in terms of their opinions/satisfaction toward the facilities. The second validity weakness in the questionnaire as a whole may have been the sheer number of questions. The participants may not have had the patience to complete the whole questionnaire with their full attention.

The focus of this study was on implementing overall resource management such that economic, aesthetic, and social requirements are all fulfilled, and due respect is paid to important aspects such as key ecological processes and biodiversity, cultural integrity, and life support systems. [35,40,41] The visitors' reactions and demands were examined through IPA analysis. The results could be used by national park managers to develop constructive suggestions on implementing sustainable development in Taiwan's tourism industries.

Possible suggestions for improving the research questionnaire are the following: (a) Decrease the number of questions to encourage more sincere replies made with full attention, (b) Apply a ranking system instead of degrees among the opinions to avoid the validity weakness from respondents choosing "no comment" or "not sure", and (c) Prepare some special offers to encourage visitors to state more opinions or expectations for future research.

**Author Contributions:** Conceptualization, H.-m.Y.; Data curation, P.J.B.T. and H.-m.Y.; Formal analysis, P.J.B.T.; Investigation, P.J.B.T.; Methodology, P.J.B.T. and H.-m.Y.; Project administration, M.-H.H.; Resources, H.-m.Y.; Visualization, M.-H.H.; Writing—original draft, P.J.B.T. All authors have read and agreed to the published version of the manuscript.

**Funding:** This research received no external funding.

**Institutional Review Board Statement:** Not applicable.

**Informed Consent Statement:** Not applicable.

**Conflicts of Interest:** The authors declare no conflict of interest.

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
