# Peer review of "A Case Study of Tourist Perceptions and Revisit Intention Regarding Qingwan Cactus Park in Penghu, Taiwan"

_sustainability, doi:10.3390/su132212404_

Round 1

Reviewer 1 Report

The article reports a study conducted in Qingwan Cactus Park to understand visitors' perception and willingness to revisit.

The study presents a thorough literature review and clearly presents the adopted methodology. 

The results are well documented and sustained and provide novel insights useful for academics, park administrators, and policymakers.

The only weak point of the study is the reference to sustainability. The authors state that the study aims at providing a useful assessment of sustainable tourism for national park sightseeing. Nevertheless, neither the results nor the discussion makes explicit reference to sustainability issues. This element should be improved in the article to better answer the initial questions.

Minor issues.

Images and graphs could be improved to increase readability and to uniform the style.

Figure 4. Quadrant diagram of IPA is not understandable without reference to what the numbers stand for.

Author Response

To Reviewer 1

  1. Thank you for your valuable comments! In the discussion, we mention that “The World Tourism Organization’s (WTO) definition was drawn from the Brundtland Report and defines sustainable tourism as the type of tourism that “meets the needs of tourists and host regions, while ensuring and enhancing future opportunities.” [50] This study analyzed and compared the differences in the surveyed visitors’ satisfaction levels prior to and after their practical experiences. We IPA (Importance0Performance Analysis) to analyze the visitors’ satisfaction levels. From the data shown in Tables 6, 7, 8, and 9, we can see that the top three features required for satisfaction included restrooms (4. 25), shaded spaces (4.17), and recreation spaces (4.02). Relatedly, Qingwan Cactus Park is located in the south of Penghu, and it has the only public restroom on the road. We were so impressed by this result, which exceeded our expectations. Furthermore, shaded spaces and recreation spaces were expected by visitors of every age. That was because of the scorching sun in summer and the windy weather in winter, which made people have the intention of finding a place for rest or shade.
  2. Images and graphs could be improved to increase readability and to uniform the style. - See Table 6, 7 and figure 4. Please see in red color.
  3. There are 24 items for Four Quadrants of IPA matrix. The numbers are as follows:

See Table 6 and Table 7, number 1 to number 24.

(2) Signage

(3)The trail beside basalt

(4)Recreation spaces

(5)Shaded spaces

(6)Restaurants

(7)Parking lots

(8)Restrooms

(9)Lakeside terraces

(10)Dream hall

(11)Greenhouse cultivation

(12)Dahsianhua place

(13)The area where animals are found

(14)The place where there are many lover locks

(15)Rock flower place

(16)Chienhuhua place

(17)Lighting facilities

(18)Medical facilities

(19)Historic sites

(20)Explanation of handbook

(21)Explanation of billboard

(22)Explanation of poster

(23)Explanation of multimedia

(24)Facilities Display

(1) Tourist visitor center

(2) Signage

(3)The trail beside basalt

(4)Recreation spaces

(5)Shaded spaces

(6)Restaurants

(7)Parking lots

(8)Restrooms

(9)Lakeside terraces

(10)Dream hall

(11)Greenhouse cultivation

(12)Dahsianhua place

(13)The area where animals are found

(14)The place where there are many lover locks

(15)Rock flower place

(16)Chienhuhua place

(17)Lighting facilities

(18)Medical facilities

(19)Historic sites

(20)Explanation of handbook

(21)Explanation of billboard

(22)Explanation of poster

(23)Explanation of multimedia

(24)Facilities Display

4.      Thank you for your wonderful comments! Your comments are highly appreciated. You did a good job.

5.      In fact, the contribution of this study is rather obvious. Please read in red (two articles).

Bynum Boley & Nancy Gard McGehee & A.L. Tom Hammett. (2017). Tourism Management Volume 58, 66-77

Jih-Kuang Chen. A New Approach for Diagonal Line Model of Importance-Performance Analysis: A Case Study of Tourist Satisfaction in China. Sage Open (2021).

Reviewer 2 Report

The research study presented is interesting, however the methodology used is very basic, therefore the conclusions as well. Furthermore, the scales used have not been validated. The document is well structured and easy to read. The methodology used to achieve the objective is very poor. The literature review is adequate and up-to-date. The objective is not very ambitious.
My recommendations and suggestions are:
1.- In the abstract: briefly include the results obtained.
2.- In the introductory section, state clearly What is the novelty of the study? What gap in literature is it intended to cover?
3.- In the introduction section: include a final paragraph indicating the sections into which the document is divided. It is also important to mention after the objective which is the methodology that will be used to achieve said objective.
4.- It is necessary to number the sections: it is difficult to identify which are sections and which are sub-sections.
5.- A model appears in Figure 3, check if it is correct, it seems that an arrow is missing (Pull factors is not linked to any other construct) and the hypotheses raised and supported in the previous literature are not.
6.- Participants: What is the sampling error?
7.- Items of the survey: put in the text "widely used survey", which one? it is necessary to quote it.
8.- Questionnaire response rate 82.6%, it is not correct, this is calculated based on the questionnaires distributed and must be based on visitors.
9.- Statistical methodology: very basic techniques (average). The scales must be valid (exploratory factor analysis) and an exploratory factor analysis should be carried out to group the items of the different scales and obtain results that allow obtaining more consistent results and conclusions.
10.- The conclusions are very basic. What are the theoretical implications? And Management implications?
11.- Expose the limitations of the investigation.

Author Response

To Reviewer 2

 1. Please see in red.

The visitors’ reactions and demands were examined through IPA analysis. The results could be used national park managers to develop constructive suggestions on implementing sustainable development in Taiwan’s tourism industries.

  1. In the introductory section, state clearly what the novelty of this study is? Is there any gap in literature that you intend to cover?

Please see in red.

In connection with this research subject, this study investigated the factors of forward expectations, practical experiences, motivations, perceptions, and revisit intentions among tourists. The following sections provide a review, discussion, and brief compendium of pertinent literature to make the theoretical basis of the study clear and better explain the overall structure of this study.

  1. In the introduction section: include a final paragraph indicating the sections into which the document is divided. It is also important to mention after the objective which is the methodology that will be used to achieve objectives.

Please see in red.

This study focused on visitors to Qingwan Cactus Park to assess whether travel expectations and tourist motivations may alter their perceptions towards the park as well as their intentions to revisit. Perceptions scales are useful tools for measuring people's views regarding products, work situations, their quality of life, community characteristics, and the quality of outdoor recreation options, among other things. In seeking to understand and satisfy the needs of tourists, it is most important to explore their motivations and perceptions levels, and in order to understand those factors, it is necessary to take appropriate measurements.

Purposes of this study were clearly presented and achieved.

Please see in red.

Understand the value of tourist perceptions and provide professional management knowledge to enhance the management quality of national park services.

Provide useful assessment of sustainable tourism for national park sightseeing.

Determine the factors that contribute to Penghu visitors’ intentions and behaviors with regard to revisiting the Cactus park.

Provide the local government with a better decision-making process.

Judge whether the satisfaction of tourists’ demands in terms of the recreational quality of national park services plays a significant role in Penghu.

Understand the value of tourist perceptions and provide professional management knowledge to enhance the management quality of national park services.

  1. It is necessary to number the sections; it is difficult to identity which sections and sub-sections are.

Thank you for your constructive suggestion! Correction was made—Section 2.1 & 2.2 & 2.3. Correction was made—Section 4. 5 is changed to Section 3.5.

  1. A model appears in Figure 3, check if it is correct, and it seems that an arrow is missing (Pull factors is not linked to any other construct) and the hypotheses raised are not supported in the previous literature.

Correction is made. An arrow is added and linked to its construct. Check Figure 3—revised.  

  1. Participants: What is the sampling error?

No error. See in red.

The participants in this study consisted of park visitors who viewed the park as a tourist attraction to be visited during a trip to the islands. More specifically, the study surveyed a total of 413 tourists who came to the park. These 413 tourists were randomly selected from among all the visitors to the park during a certain period and included individuals from a range of different cities.

IPA research should include semi- and non-structured interviews, where between two (2) and twenty five (25) individuals participate.

  1. Items of the survey: put in the text “widely used survey”, which one? It is necessary to quote it.

Please see references [70] and [71]

[70] Phau, Ian & Lee, Sean & Quintal, Vanessa. (2013). An investigation of push and pull motivations of visitors to private parks The case of Araluen Botanic Park. Journal of Vacation Marketing. 19. 269-284. 10.1177/1356766712471232.

[71] Bynum Boley & Nancy Gard McGehee & A.L. Tom Hammett. (2017). Tourism Management. Volume 58, 66-77.

  1. Questionnaire response rate 82.6%, it is not correct, this is calculated based on the questionnaires distributed and must be based on visitors.

Thank you for your comments!

  1. Statistical methodology: very basic techniques (average). The scales must be valid (exploratory factor analysis) and an exploratory factor analysis should be carried out to group the items of the different scales and obtain results that allow obtaining more consistent results and conclusions.

IPA research should include semi- and non-structured interviews, where between two (2) and twenty five (25) individuals participate.

Please read Bynum Boley & Nancy Gard McGehee & A.L. Tom Hammett. (2017). Tourism Management. Volume 58, 66-77.  See in red.

With respect to research population and sample size, Creswell (2012) [30] explained that determining the size of the sample that will be required is vital when selecting study participants. Similarly, a typical phenomenological study can have a participant sample size of between 2 and 25. Participants in an IPA research study should also be drawn from a homogeneous sample pool, for which Smith et al. (2009) [32] pointed out the main challenge in IPA is the following: “IPA is a detailed account of individual experience. The issue is quality, not quantity.

Creswell’s (2013) [31] asserted that the exploratory capacity in qualitative research enables investigating, interpreting, and understanding the problems encountered in these studies. Additionally, he claimed that “We conduct qualitative research because a problem or issue needs to be explored”; therefore, he was of the opinion that the phenomenological approach is most suitable for uncovering the root-cause of a phenomenon.

  1. The conclusions are very basic. What are the theoretical implications? And Management implications?

They are not basic at all. They are very important because of the following reasons. See in red.

Our study is a source for national park managers, policymakers, along with communication strategists, since it presents crucial practical and managerial implications. These should be taken into consideration to better understand how push and pull factors can influence the intentions of tourists to revisit a destination, as well as the behavioral processes behind choosing a national park as a destination.

  1. Expose the limitations of the investigation

Please see in red. .

a.) Decrease the number of questions to encourage more sincere replies made with full attention, b.) Apply a ranking system instead of degrees among the opinions to avoid the validity weakness from respondents choosing "no comment" or "not sure", and c.) Prepare some special offers to encourage visitors to state more opinions or expectations for future research. 

Reviewer 3 Report

Dear Authors

This is an interesting topic, but I have some comments and some questions. The article needs to be revised according to the structure guidelines. There are too many single parts in the first chapter, which makes this chapter unreadable. The same applies to chapter 4. In addition, the discussion of the results should be expanded to include the results of other or similar studies.

Questions:

When was the study conducted and what is the impact of the covid pandemic?

How was the sample size determined?

How were individuals selected for the study?

What nationality are the study participants?

Of how many parts, how many questions did the questionnaire consist of?

Is this sample size appropriate for a study based on this questionnaire?

What is the content of section 2.2?

What are the conclusions and practical recommendations?

Author Response

To Reviewer 3

  1. When was the study conducted and what is the impact of Covid pandemic?

Please see in red.

The developed questionnaire was distributed to the randomly selected visitors to the park from August 2019 to September 2019. In other words, the study’s measurements of tourists’ perception came from data collected from tourists when they were still at the park. This approach was taken because it meant that the tourists would be in the midst of experiencing their visits and, thus, would have fresh and clear opinions regarding the facilities, attractions, and customer services of the park. The total number of questionnaires distributed was 500, of which a total of 413 returned questionnaires were valid, which represents an effective rate of 82.6%. The questionnaire was intended to measure the visitors’ levels of satisfaction with various aspects of the different facilities/areas in the park.

  1. How was the sample size determined?

Sample size is not an important factor in qualitative research method.

  1. How were individuals selected for the study?

Please see in red.

A phenomenological research study conducted by Creswell (2013) [31] revealed that collecting information is a process that entails primary in-depth interviews with up to 10 participants, in which the key is to describe the essence of the phenomenon for the few who have experienced it. Therefore, collecting data for an IPA study should proceed following these instructions: IPA research should include semi- and non-structured interviews, where between two (2) and twenty five (25) individuals participate.

  1. What nationality are the study participants?

Taiwan citizens, local residents at Penghu, and only a few foreigners. A total of 413 returned questionnaires were valid, which represents an effective rate of 82.6%. The questionnaire was intended to measure the visitors’ levels of satisfaction with various aspects of the different facilities/areas in the park.

  1. Of how many parts, how many questions did the questionnaire consist of?

Please refer to the questionnaire.

  1. Is this sample size appropriate for a study based on this questionnaire?

Yes, it is appropriate. Sample size is not an important factor in qualitative research method

  1. What is the content of section 2.2?

2.2 is not missing and it has been reordered.

 What are the conclusions and practical recommendations?

  1. Please see in red.

Our study is a source for national park managers, policymakers, along with communication strategists, since it presents crucial practical and managerial implications. These should be taken into consideration to better understand how push and pull factors can influence the intentions of tourists to revisit a destination, as well as the behavioral processes behind choosing a national park as a destination.

Reviewer 4 Report

The work is interesting, but the current version is not publishable because it needs a lot of improvements, I will detail some recommendations that the authors should make to improve the work:

1. The authors should clarify and focus the objectives of the work, at the end of page 2 they cite 5. And then on page 10 they indicate that the objective of the work is:
The current study's main purpose was to investigate the effects of tourists' expectations and experiences on their levels of satisfaction with the Qing wan Cactus Park in Penghu.

And in the abstract they state:

The purpose of this study was to examine factors influencing tourists' visits to Qingwan Cactus Park, including their reasons for visiting and their perceptions of their visits. 

It is really necessary so much incoherence, they are similar but not the same.

Regarding the introduction, it describes sustainability by talking about the three dimensions of the Triple Bottom Line Theory, but it does not allude to it anywhere.

The literature review is not connected to the research objectives, whatever the central one is.

The IPA analysis methodology is suitable for groups of 10 or less individuals, as indicated by the authors, so what are the advantages of applying it to a survey of more than 400 individuals. It is not coherent. The IPA analysis that is indicated in the paper is impossible to apply, as it is impossible as indicated. The analysis of results is poor and does not connect with either theory or practice.

The contribution of the work is poor in the current version, the last section does not explain how this work advances in the line of previous ones, it does not have a strong theoretical support, and the results are vague and excessively descriptive. The survey was not conducted for academic purposes, but simply as a data collection instrument.

The work does not pose hypotheses, nor research question, it does not seem academic, nor suitable for a Q1 JCR journal.

Finally,  Authors should present more deeply the theoretical gap. How the findings of this paper influence on managerial decisions? Results are presenting in a positive manner, no limitations or constraints are considered.

Author Response

To Reviewer 4

Thank you for your precious comments! Authors did not mention the three dimensions of the Triple Bottom Line Theory. We focus on push and pull motivation model and market segmentation theory. Please see Table 3. We wonder whether you are familiar with this model. Two types of importance-performance analysis (IPA), Interpretative phenomenology analysis (IPA) and quadrant model were applied widely in tourism industry. The first IPA usually is adopted in service industry while the other is used in applied psychology. The first IPA is a popular tool because it is easy to operate and its results are easily interpretable. IPA was successful applied to determine a management strategy of tourist satisfaction. Please read “A New Approach for Diagonal Line Model of Importance-Performance Analysis: A Case Study of Tourist Satisfaction in China” by Jih-Kuang Chen—Sage Open (2021) and “Importance-performance analysis (IPA) of sustainable tourism initiatives: The resident perspective” by Bynum Boley, Nancy Gard McGehee, A.L. Tom Hammett —Tourism Management (2017).

  1. Dear Reviewer:

We would appreciate it if you can re-examine your English grammar before sending out your comments. To be honest, your English writing is very Chinese. Please accept my apology.

  1. Please see in red.

In addition, due to its qualitative nature, IPA constitutes an ideal opportunity for researchers to better acquaint themselves with the innermost deliberation of “lived experiences” from interviewees (study participants). Being a “participant-oriented” approach, the interpretative phenomenological analysis approach makes it possible for interviewees to express themselves and convey their “lived experience” stories in a manner that they deemed to be appropriate and in the absence of any distortion and/or prosecution. Smith et al. (2009) [32] established that the field of psychology gave origin to IPA, as well as a considerable number of previous studies that utilized IPA were health psychology studies. Since then, IPA has seen increasingly frequent use, particularly in clinical and counseling psychology and social and educational psychology.

  1. Please see in red.

With respect to research population and sample size, Creswell (2012) [30] explained that determining the size of the sample that will be required is vital when selecting study participants. Similarly, a typical phenomenological study can have a participant sample size of between 2 and 25. Participants in an IPA research study should also be drawn from a homogeneous sample pool, for which Smith et al. (2009) [32] pointed out the main challenge in IPA is the following: “IPA is a detailed account of individual experience. The issue is quality, not quantity.

  1. Please see in red.

Creswell’s (2013) [31] asserted that the exploratory capacity in qualitative research enables investigating, interpreting, and understanding the problems encountered in these studies. Additionally, he claimed that “We conduct qualitative research because a problem or issue needs to be explored”; therefore, he was of the opinion that the phenomenological approach is most suitable for uncovering the root-cause of a phenomenon.

Round 2

Reviewer 3 Report

Dear Authors
This article has been significantly revised in accordance with your suggestions.

Reviewer 4 Report

Dear Authors

I hope that you understand this:

Which is the theoretical contribution of the manuscript?

Minor changes:

Use IPA for two different concepts could confuse the readers.

regards,

This manuscript is a resubmission of an earlier submission. The following is a list of the peer review reports and author responses from that submission.

Round 1

Reviewer 1 Report

The manuscript must be improved considerably in its design. Tables are using most of the space of the paper. These tables must be shortened and fitted to one page (Tables 4, 5, 6, 7 and 9). It exists the possibility to place some of them in the annexe. 

It is crucial, as well, the inclusion in the bibliographic framework, of some references to sustainable tourism's psychologist dimension. It can be used for it, the next references:

Álvarez-Sousa, A. (2018). The Problems of Tourist Sustainability in Cultural Cities: Socio-Political Perceptions and Interests Management. Sustainability, 10(2), 503. https://doi.org/10.3390/su10020503

Butler, R. (1980). The Concept of a Tourist Area Cycle of Evolution: Implications for Management of Resources. Canadian Geographer. Le Géographe Canadien, 24, 5–12. https://doi.org/10.1111/j.1541-0064.1980.tb00970.x

Pérez de las Heras, M. (2004): Manual del Turismo sostenible. Edit. Mundi-Prensa. Madrid.

World Tourism Organization (2005). Indicators of Sustainable Development for Tourism Destinations A Guidebook.

Finally, a minor consideration is about the description and justification of the study surveyed. 

Author Response

Comments and Suggestions for reviewer’s 1

The manuscript must be improved considerably in its design. Tables are using most of the space of the paper. These tables must be shortened and fitted to one page (Tables 4, 5, 6, 7, and 9). It exists the possibility to place some of them in the annexed. -----------modified-ok

It is crucial, as well, the inclusion in the bibliographic framework, of some references to sustainable tourism's psychologist dimension. It can be used for it, the next references:

Álvarez-Sousa, A. (2018). The Problems of Tourist Sustainability in Cultural Cities: Socio-Political Perceptions and Interests Management. Sustainability, 10(2), 503. -Revised--page 9

Butler, R. (1980). The Concept of a Tourist Area Cycle of Evolution: Implications for Management of Resources. Canadian Geographer. Le Géographe Canadien, 24, 5–12. https://doi.org/10.1111/j.1541-0064.1980.tb00970.x---------Revised-10

Pérez de las Heras, M. (2004): Manual del Turismo sostenible. Edit. Mundi-Prensa. Madrid.---- -Revised and modified—p 11

World Tourism Organization (2005). Indicators of Sustainable Development for Tourism Destinations A Guidebook.-------   modified

Finally, a minor consideration is about the description and justification of the study surveyed. 

-------------------------------Revised

Reviewer 2 Report

The research topic is interesting but does not bring anything new in the area of satisfaction research with tourist attractions. The results may only be useful to Park managers. The article is not adapted to the MDPI editing requirements - it requires a significant modification. The purpose of the research is unclear and inconsistent. Some parts of the article are chaotic and disordered. In the article, there are stylistic and linguistic errors.

Detailed comments:

Abstract:

  1. The most important conclusions are missing.
  2. There is no description of the methods used in the article (e.g., IPA).
  3. The purpose of the article is missing.

Introduction

  1. The paragraph on sustainable development is redundant and does not apply to the article's subject.
  2. The purpose of the study is missing.
  3. The paragraph that describing the structure of the paper is missing.

Literature Review:

  1. The literature review is too modest.
  2. The titles of tables and figures are missing.

Methodology:

  1. There is a lack of research review and tools for measuring the satisfaction of tourist attractions.
  2. There is no justification for selecting the dimensions of satisfaction.
  3. How was the randomness of the research sample ensured?
  4. Please include the questionnaire as an attachment.

Results and interpretations:

  1. Figure 3 is not legible.
  2. SWOT analysis is redundant.
  3. Please describe the points of the measurement scale of tourists' motivation.
  4. Tables require significant modifications - there is no need to include points for measurement scales for each item.
  5. There is no assessment of the measurement reliability of the applied tool.
  6. There is no justification for the application of the IPA method. Are there any alternative methods? What makes the IPA method advantageous?

Discussion and conclusion:

  1. How do the obtained results compare to other studies?
  2. There is no information about the revision of the hypothesis.
  3. The conclusions of the survey are a bit trivial. The justification for the need to modify measurement tools should be based, among others, on the results of a "solid" statistical analysis.

Author Response

Comments and Suggestions for   reviewer’s 2

The research topic is interesting but does not bring anything new in the area of satisfaction research with tourist attractions. The results may only be useful to Park managers. The article is not adapted to the MDPI editing requirements - it requires a significant modification. The purpose of the research is unclear and inconsistent. Some parts of the article are chaotic and disordered. In the article, there are stylistic and linguistic errors.

Detailed comments:

Abstract: Revised and modified   light green

  1. The most important conclusions are missing --------p18,19',20
  2. There is no description of the methods used in the article (e.g., IPA).---p10,11
  3. The purpose of the article is missing.------p2

Introduction------ Revised and modified p1,p2,p8,p9

  1. The paragraph on sustainable development is redundant and does not apply to the article's subject.
  2. The purpose of the study is missing.
  3. The paragraph that describing the structure of the paper is missing.

Literature Review: Revised and modified p8,p9,p10

  1. The literature review is too modest.
  2. The titles of tables and figures are missing.

Methodology: Revised and modifiedp8,p9,p10,p25-28

  1. There is a lack of research review and tools for measuring the satisfaction of tourist attractions.
  2. There is no justification for selecting the dimensions of satisfaction.
  3. How was the randomness of the research sample ensured?
  4. Please include the questionnaire as an attachment.

Results and interpretations: Revised and modified

  1. Figure 3 is not legible.---------removed
  2. SWOT analysis is redundant.----- removed
  3. Please describe the points of the measurement scale of tourists' motivation.----ok
  4. Tables require significant modifications - there is no need to include points for measurement scales for each item.----ok
  5. There is no assessment of the measurement reliability of the applied tool---IPA.
  6. There is no justification for the application of the IPA method. Are there any alternative methods? What makes the IPA method advantageous?    Ok   P10-11

Discussion and conclusion: Revised and modified

  1. How do the obtained results compare to other studies?  ok
  2. There is no information about the revision of the hypothesis.ok
  3. The conclusions of the survey are a bit trivial. The justification for the need to modify measurement tools should be based, among others, on the results of a "solid" statistical analysis.ok

Reviewer 3 Report

Dear Authors, 

the idea and the article are interesting, but insufficiently developed and require in-depth research still.

In the theoretical part, outdated literature prevails. In addition, the authors' understanding of many concepts is incorrect.

"The process of recreational experience" - concerns tourist experience - not recreational - because it contains the travel phase. So it is not a recreational satisfaction but tourists satisfaction.

Figure 2 - image title is missing, Table 1. - title is missing.

Crompton (1979) writes about pull and push motivations - not thrust.

Table 1 presents a horrible old source from 1964! Why particulary this one? In general, the authors do not discuss current sources.

The authors confuse the concepts of the perception and the satisfaction. The IPA analyzes expectations and perceptions, not satisfaction. Satisfaction is only calculated from the difference between perception and satisfaction (what the authors wrote in the paper).

What is it "forward expectetions', "practical experiences"?

Why the authors did not calculate the relationship between motivation, expectations, perception and behavioral intentions? This is the article's great weakness and it must necessarily be done in order to identify the factors that influence the intentions.

What was figure 3 placed for? This has nothing to do with the purpose of the research. Similarly, the SWOT analysis (4.2) - what for?

Table 4 is redundant. The results in the following tables are presented incorrectly - see how other authors do it.

4.5. - Practical experiences - this is a perception - not satisfaction.

Table 9. - Overall satisfaction - it is not revisit intentions.

Author Response

Comments and Suggestions for reviewer 3

Dear Authors, 

the idea and the article are interesting, but insufficiently developed and require in-depth research still. Modified and revised-----p10-p11

In the theoretical part, outdated literature prevails. In addition, the authors' understanding of many concepts is incorrect. Modified and revised  p 8,9,10,11

"The process of recreational experience" - concerns tourist experience - not recreational - because it contains the travel phase. So it is not a recreational satisfaction but tourists satisfaction. Modified and revised

Figure 2 - image title is missing, Table 1. - title is missing.------ Modified and revised

Crompton (1979) writes about pull and push motivations - not thrust. Modified and revised

Table 1 presents a horrible old source from 1964! Why particulary this one? In general, the authors do not discuss current sources. Modified and revised

The authors confuse the concepts of the perception and the satisfaction. The IPA analyzes expectations and perceptions, not satisfaction. Satisfaction is only calculated from the difference between perception and satisfaction (what the authors wrote in the paper).--- Modified and revised

What is it "forward expectetions', "practical experiences"?--- Modified and revised

Why the authors did not calculate the relationship between motivation, expectations, perception and behavioral intentions? This is the article's great weakness and it must necessarily be done in order to identify the factors that influence the intentions. Modified and revised

What was figure 3 placed for? This has nothing to do with the purpose of the research. Similarly, the SWOT analysis (4.2) - what for? Removed ,Modified and revised

Table 4 is redundant. The results in the following tables are presented incorrectly - see how other authors do it. Removed ,Modified and revised

4.5. - Practical experiences - this is a perception - not satisfaction. Removed ,Modified and revised

Table 9. - Overall satisfaction - it is not revisit intentions. Removed ,Modified and revised

Round 2

Reviewer 1 Report

The author/s must improve the design of the manuscript. It does not exist paragraphs between tables 4, 5, 6 and 7, and Figure 3. Esthetically it is not appropriate.  Probably, it must be extended the comments of these tables and figures. 

Author Response

Dear Reviewer 1

  Thank you very much for your valuable comments. We have revised the manuscript based on your request.

   Thank you very much.

 Best regards

      AUTHOR(s)

Dear Reviewer1,2, 3, and Editor

  Thank you very much for your valuable comments. We have revised the manuscript based on your request. (light green,  yellow, red, and regular blue-figure 3)
   Thank you very much.
 Best regards

      AUTHOR(s)

Reviewer 2 Report

I would like to thank you for the corrections made. Unfortunately, the article is still disordered and difficult to read. The article is still not adapted to the MDPI editing requirements - it requires a significant modification.

Author Response

Dear Reviewer 2

  Thank you very much for your valuable comments. We have revised the manuscript based on your request.

   Thank you very much.

 Best regards

      AUTHOR(s)

Dear Reviewer1,2, 3, and Editor

  Thank you very much for your valuable comments. We have revised the manuscript based on your request. (light green,  yellow, red, and regular blue-figure 3)
   Thank you very much.
 Best regards

      AUTHOR(s)

Reviewer 3 Report

Dear authors, thank you for significantly improving the submitted text, but it is still of very low quality.

The title of the article still doesn't reflect the problem.

Why were motivations studied? How do they relate to expectations, perceptions, satisfaction, and intentions?

Where in the article are the results of principal component analysis about which the authors write in the abstract?

Where in the article are the results of the research reactions and demands? - see the abstract

Language: "The results could be used national park managers..."

Citation method - Authors are still not following the recommended citation method.

Repeating the same text: "Purposes of this study" and again in "Methodology".

Table 3 and 4 are identical - the motivations.

What is the relationship of expectations and perceptions with satisfaction and behavioral intentions?

The authors do not answer these and many other questions in the article. 

Author Response

Dear Reviewer 3

  Thank you very much for your valuable comments. We have revised the manuscript based on your request.

   Thank you very much.

 Best regards

      AUTHOR(s)

Dear Reviewer1,2, 3, and Editor

  Thank you very much for your valuable comments. We have revised the manuscript based on your request. (light green,  yellow, red, and regular blue-figure 3)
   Thank you very much.
 Best regards

      AUTHOR(s)

Round 3

Reviewer 2 Report

Dear Authors, 

Thank you for the corrections made, but they did not improve the article enough. Unfortunately, the article does not meet the requirements of the publication in the Sustainability.

Best Regards

Reviewer 3 Report

Sorry, but the article is of very poor quality in all respects:

  • editing,
    - logical,
    - scientific,
    - practical value.

I am deeply convinced that the article is not suitable for publication.